# Minimal biophysical model of combined antibiotic action

**Bor Kavčič**[1], **Gašper Tkačik**[1], **Tobias Bollenbach**[2,3]*

**1** Institute of Science and Technology Austria, Klosterneuburg, Austria, **2** Institute for Biological Physics, University of Cologne, Cologne, Germany, **3** Center for Data and Simulation Science, University of Cologne, Cologne, Germany

* t.bollenbach@uni-koeln.de

## Abstract

Phenomenological relations such as Ohm's or Fourier's law have a venerable history in physics but are still scarce in biology. This situation restrains predictive theory. Here, we build on bacterial "growth laws," which capture physiological feedback between translation and cell growth, to construct a minimal biophysical model for the combined action of ribosome-targeting antibiotics. Our model predicts drug interactions like antagonism or synergy solely from responses to individual drugs. We provide analytical results for limiting cases, which agree well with numerical results. We systematically refine the model by including direct physical interactions of different antibiotics on the ribosome. In a limiting case, our model provides a mechanistic underpinning for recent predictions of higher-order interactions that were derived using entropy maximization. We further refine the model to include the effects of antibiotics that mimic starvation and the presence of resistance genes. We describe the impact of a starvation-mimicking antibiotic on drug interactions analytically and verify it experimentally. Our extended model suggests a change in the type of drug interaction that depends on the strength of resistance, which challenges established rescaling paradigms. We experimentally show that the presence of unregulated resistance genes can lead to altered drug interaction, which agrees with the prediction of the model. While minimal, the model is readily adaptable and opens the door to predicting interactions of second and higher-order in a broad range of biological systems.

## Author summary

Applying multiple antibiotics simultaneously can boost treatment effectiveness and aid against rampant antibiotic resistance. Because of the impractically large number of possible combinations of drugs, those that are effective are found by trial and error. Hence, a predictive theory to characterize drug cocktails would be of enormous value. Recently identified phenomenological laws ease the construction of predictive models of bacterial growth. Here, we build a model of the effects of antibiotic combinations on bacteria and show that it makes reliable predictions for experimental outcomes. Our model takes responses to individual drugs as inputs and predicts their combined effect. This output determines the type of drug interaction, which can range from antagonistic (the combined

**Data Availability Statement:** All relevant data are available at https://doi.org/10.15479/AT:ISTA:8930.

**Funding:** This work was supported in part by Tum stipend of Knafelj foundation (to B.K.), Austrian Science Fund (FWF) standalone grants P 27201-B22 (to T.B.) and P 28844(to G.T.), HFSP program

Grant RGP0042/2013 (to T.B.), German Research Foundation (DFG) individual grant BO 3502/2-1 (to T.B.), and German Research Foundation (DFG) Collaborative Research Centre (SFB) 1310 (to T.B.). The funders had no role in study design, data collection and analysis, decision to publish, or preparation of the manuscript.

**Competing interests:** The authors have declared that no competing interests exist.

effect is weaker) to synergistic (the combined effect is stronger). We broaden the model by including the direct physical interaction on the target, drug resistance genes that alter the drug interaction, and drugs that mimic poor growth environments by choking the supply of growth-essential components, which we test experimentally. Our results prove how biophysical models that use empirical laws can predict responses to drug combinations. Importantly, such models can successfully predict mechanisms underlying interactions of drug combinations. This approach is extensible to combinations of more than two drugs and diverse biological systems.

## Introduction

Antibiotics are small molecules that interfere with essential processes in bacterial cells, thereby inhibiting growth or even killing bacteria [1]. Even though antibiotics have been used in the clinic for nearly a century and have molecular targets that are often known, bacterial responses to antibiotics are complex and largely unpredictable. Due to the looming antibiotic-resistance crisis [2], understanding and predicting bacterial responses to antibiotics is becoming increasingly important. A promising way to fight the emergence and spread of resistant pathogens is to use more than one drug simultaneously [3]; however, predicting the effects of such drug combinations is a great challenge.

The combined effect of antibiotics emerges from the complex interplay of individual drug effects and the physiological response of the cell to the drug combination. Drug interactions are determined by the combined effect of multiple drugs on cell growth and survival. These interactions are defined with respect to an additive reference. By definition, additive drugs act as substitutes for each other; synergy occurs if the combined effect of the drugs is stronger than in the additive reference case, and antagonism occurs if the combined effect is weaker. An extreme case of antagonism–termed suppression–occurs when one of the drugs loses its potency in the presence of the other drug, *i.e.*, bacterial growth is accelerated by adding one of the drugs. In practice, such drug interactions are determined by measuring bacterial growth in a two-dimensional assay in which each drug is dosed in a gradient along each axis.

By measuring the growth rate $\lambda$ over a two-dimensional matrix of drug concentrations $(c_A, c_B)$, the dose-response surface $y(c_A, c_B) = \lambda(c_A, c_B)/\lambda_0$ is obtained; here, $\lambda_0$ is the growth rate in the absence of antibiotics. The dose-response surface can be characterized by the shape of its contours, *i.e.*, the lines of equal growth rate (isoboles). By definition, in the additive case these contours are linear (but not necessarily parallel) (Fig 1). Linear isoboles imply that increasing the concentration of one drug is equivalent to increasing the concentration of the other. Drug interactions according to the Loewe definition [4] occur when the dose-response surface deviates from this additive expectation. The interaction is synergistic or antagonistic if the drug combination is more or less potent than the additive expectation, respectively (Fig 1). There are other definitions of drug interactions: In the Bliss definition, the responses to individual drugs are multiplied to yield the reference response to the combination [5, 6]. The definition of Bliss independence is intuitive and simple to evaluate; yet, it generally fails to recognize that the same drug should exhibit additivity, *i.e.*, the combined effect of the drug with itself should be $y(a + b) = y(c + d)$, if $a + b = c + d$. If this is the case, then for Bliss independence $y(a + b) = y(a) \times y(b)$–this relation holds only if the dose-response curve is exponential. Therefore, it is not possible to recognize additive interactions using Bliss independence for drugs with non-exponential dose-response curve. In this work, we use the Loewe definition because a predictive model of drug interactions has to produce an additive surface if a drug is

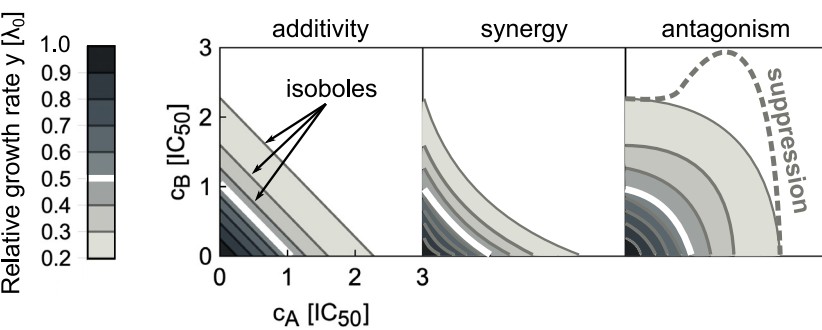

**Fig 1. Drug interaction types are defined by the shape of the dose-response surface.** The dose-response surface is given by the relative growth rate $y$ as a function of the concentrations of two drugs ($c_A$, $c_B$). Here, the concentrations are normalized such that $c_i = 1$ corresponds to 50% growth inhibition in the presence of drug $i$ alone. The dose-response surface is defined by lines of constant growth rate (isoboles); the white isobole corresponds to 50% growth inhibition; different shades of gray correspond to different relative growth rates. Additivity (left) represents the case in which antibiotics act as substitutes for each other. Synergy (middle) and antagonism (right) are characterized by convex or concave isoboles (curving towards or away from the origin), respectively. Isoboles corresponding to suppression (dashed gray) are non-monotonic; in the example shown, this implies that adding drug A on top of drug B increases the growth rate.

combined with itself. However, we are principally interested in the shape of entire dose-response surfaces, which is independent of the exact definition of drug interactions.

Higher-order interactions, which occur when more than two drugs are combined, can be predicted to some extent using mechanism-independent models [7, 8]. However, such predictions generally require prior knowledge of the pairwise interactions between all drugs involved, which can usually not be predicted. One of the main reasons for this situation is that the underlying mechanisms of drug interactions are largely unknown. Systematic measurements of all pairwise drug interactions are hampered by the combinatorial explosion of possible drug-drug and concentration-concentration combinations, which prohibit such brute-force approaches. Thus, to guide the design and analysis of drug combinations, it is crucial to develop predictive theoretical models.

Apart from their clinical importance, antibiotics targeting the bacterial ribosome (translation inhibitors) are particularly well-suited for biophysical modeling since the physiological response to perturbations of translation can be described quantitatively using bacterial growth laws [9, 10]. These empirical relations offer a phenomenological description of the growth-dependent state of the bacterial cell and provide a solid foundation for quantitative studies of bacterial physiology. Similar to laws in physics, such as Fourier's law of heat conduction or Ohm's law, these phenomenological relations enable the construction of predictive mathematical models without free parameters even if their microscopic origins are not yet understood [11]. Translation inhibitors have the additional advantage that many of the drug interactions occurring between them have been recently measured [12]. In Ref. [12] we introduced a biophysical model that we only used in a particularly simple limiting case to provide a null expectation for the measured drug interactions. While predictive, the behavior of the model outside the chosen limit, its broader implications, and possible extensions were left unexplored.

Here, we present a complete analysis of the biophysical model that predicts bacterial growth responses to combinations of translation inhibitors and its non-trivial theoretical predictions. Starting from responses to single antibiotics, we derive approximate analytical solutions of this model and investigate the effects of direct physical or allosteric interactions between antibiotics on the ribosome. We discuss several relevant extensions of the model, in particular (1) interactions with antibiotics that induce starvation, (2) the effects of resistance genes, (3) the

correspondence to non-mechanistic models of interactions between more than two drugs, and (4) predictions for interactions of translation inhibitors with antibiotics that alter growth law parameters. We validate several non-trivial predictions made by the biophysical model in experiments.

## Results

### Model for a single translation inhibitor

First, we recapitulate the biophysical model for a single translation inhibitor [10]. The model captures the kinetics of antibiotic transport into the cell and binding to the ribosome (Fig 2A and 2B), as well as the physiological response of the cell to translation perturbation. This physiological response is described by bacterial growth laws, which summarize the interdependence of the intracellular ribosome concentration $r$ and the growth rate $\lambda$ (Fig 2B). Bacterial physiology and the response to antibiotic treatment strongly depend on the nutrient environment. In particular, the number of ribosomes per cell varies over approximately $5 - 75 \times 10^3$ [13]. The ribosome concentration increases linearly with the growth rate when the latter is varied by

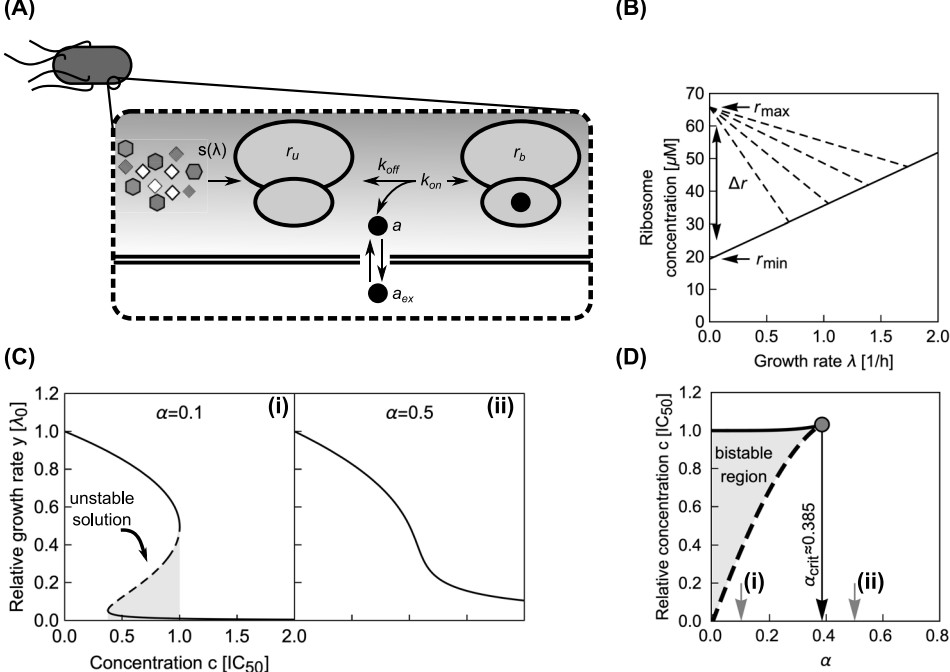

**Fig 2. Main components of the model for a single translation inhibitor and exemplary dose-response curves.**
(A) Schematic of processes captured by the model. Ribosomes (double ovals) are synthesized with the rate $s(\lambda)$ and are initially unbound by antibiotic ($r_u$). Unbound ribosomes contribute to growth. Antibiotics enter the cell ($a_{ex} \to a$) and bind to and detach from ribosomes with second-order and first-order rate constants $k_{on}$ and $k_{off}$, respectively. Bound ribosomes ($r_b$) do not contribute to growth [10]. (B) Bacterial growth laws. When the growth rate is varied by changing the quality of the nutrient environment, the ribosome concentration increases linearly with growth rate (solid line). If growth is inhibited by a translation inhibitor, the ribosome concentration increases with decreasing growth rate (dashed lines). The intercepts of the solid and dashed lines determine the minimal ($r_{min}$) and maximal ribosome concentration ($r_{max}$), respectively, which are $\Delta r$ apart [9, 10]. (C) Examples of dose-response curves. The model can produce both (i) steep and (ii) shallow dose-response curves, depending on the parameter $\alpha$ (see text). The steep dose-response curve has a region of concentrations (gray shaded area) where one unstable (dashed line) and two stable (solid lines) solutions exist. (D) Exact phase diagram for dose-response curves. The shaded area shows the region of drug concentrations where two stable solutions exist. Gray arrows show $\alpha$ for the examples from (C); black arrow shows the critical value $\alpha_{crit} = 2/3\sqrt{3}$ above which no bistability can occur.

changing the quality of the growth medium:

$$r_u = r_{\min} + \lambda/\kappa_t, \tag{1}$$

where $\kappa_t = 0.06\ \mu M^{-1} h^{-1}$, $r_u$ and $r_{\min} = 19.3\ \mu M$ are the translational capacity, the concentration of unperturbed ribosomes, and a minimal ribosome concentration, respectively [9, 10]. This first growth law states that unperturbed ribosomes synthesize new proteins, whose overall synthesis rate is proportional to the growth rate. This relation holds across diverse growth media and different *Escherichia coli* strains [9]. Typical values for doubling times range from hours to approximately twenty minutes, corresponding to growth rates up to around $2.5\ h^{-1}$. However, when the growth rate is lowered by addition of a translation inhibitor in a constant nutrient environment, the total ribosome concentration $r_{\text{tot}}$ and growth rate become negatively correlated [9]. Mathematically, this dependency is given as:

$$r_{\text{tot}} = r_u + r_b = r_{\max} - \lambda \Delta r [1/\lambda_0 - 1/(\kappa_t \Delta r)], \tag{2}$$

where $r_{\max} = 65.8\ \mu M$ is the maximal ribosome concentration, $\Delta r = r_{\max} - r_{\min} = 46.5\ \mu M$ is the dynamic range of ribosome concentration, $r_b$ is the concentration of antibiotic-bound ribosomes, and $\lambda_0$ is the maximal growth rate in the absence of antibiotics [9]. Eq (2) quantitatively describes the upregulation of ribosome production that occurs in response to translation inhibition: Bacteria produce more ribosomes to compensate for the ribosomes blocked by antibiotics.

When antibiotics enter the cell, they can bind to ribosomes. The net rate of forward and reverse binding of antibiotics to the ribosome is given by $f(r_u, r_b, a) = -k_{\text{on}}\, a(r_u - r_{\min}) + k_{\text{off}}\, r_b$, where $k_{\text{off}}$ and $k_{\text{on}}$ are first and second order rate constants, and $a$ is the intracellular antibiotic concentration (Fig 2A). Here, we assumed that only ribosomes capable of translation ($r_u - r_{\min}$) can be bound by the antibiotics [10, 14].

The intracellular antibiotic concentration is affected by the kinetics of antibiotic entry into the cell, which is given by $J(a_{\text{ex}}, a) = p_{\text{in}} a_{\text{ex}} - p_{\text{out}}\, a$, where $a_{\text{ex}}$ is the extracellular antibiotic concentration. Typical influx and efflux rates, $p_{\text{in}}$ and $p_{\text{out}}$, for different translation inhibitors range from $1 - 1000\ h^{-1}$ and from $0.01 - 100\ h^{-1}$, respectively. Typical rates of forward and reverse binding, $k_{\text{on}}$ and $k_{\text{off}}$, are around $1000\ \mu M^{-1} h^{-1}$ and between $0 - 10^5\ h^{-1}$, respectively [10, 14]. Here, $k_{\text{off}} = 0$ corresponds to antibiotics with effectively irreversible binding such as streptomycin [10, 15]. All molecular species in the cell are effectively diluted at rate $\lambda$ as cells grow and divide. Since the ribosome concentration is determined by Eq (2), the ribosome synthesis rate $s$ depends on the growth rate, *i.e.*, $s = s(\lambda)$. Together, these terms constitute a closed system of ordinary differential equations:

$$\frac{da}{dt} = -\lambda a + f(r_u, r_b, a) + J(a_{\text{ex}}, a), \tag{3a}$$

$$\frac{dr_u}{dt} = -\lambda r_u + f(r_u, r_b, a) + s(\lambda), \tag{3b}$$

$$\frac{dr_b}{dt} = -\lambda r_b - f(r_u, r_b, a). \tag{3c}$$

In a steady state of exponential growth, the ribosome synthesis rate reads $s(\lambda) = \lambda r_{\text{tot}} = \lambda \{r_{\max} - \lambda \Delta r [1/\lambda_0 - 1/(\kappa_t \Delta r)]\}$; we follow Ref. [14] by using this expression to describe ribosome synthesis for cases in which $d\lambda/dt \neq 0$ as well. The steady-state solution of Eq (3) represents a balanced-growth state of the system–the situation that is commonly

investigated in experiments. The steady-state solution reads [10]

$$0 = \left(\frac{\lambda}{\lambda_0}\right)^3 - \left(\frac{\lambda}{\lambda_0}\right)^2 +$$
$$+ \left(\frac{\lambda}{\lambda_0}\right)\left[\frac{1}{4}\left(\frac{\lambda_0^*}{\lambda_0}\right)^2 + \frac{a_{\text{ex}}}{2\text{IC}_{50}^*}\left(\frac{\lambda_0^*}{\lambda_0}\right)\right] - \frac{1}{4}\left(\frac{\lambda_0^*}{\lambda_0}\right)^2,$$

$$(4)$$

where $\lambda_0^* = 2\sqrt{p_{\text{out}}\kappa_t K_D}$ with $K_D = k_{\text{off}}/k_{\text{on}}$, and $\text{IC}_{50}^* = \Delta r \lambda_0^*/2p_{\text{in}}$. We can recast Eq (4) into:

$$\frac{1}{\alpha^2 + 1}\left(\frac{\alpha^2}{y} - \alpha^2 + 4y - 4y^2\right) - c = 0$$

$$(5)$$

by defining $a_{\text{ex}} = c \times \text{IC}_{50}$, $\lambda = y \times \lambda_0$ and $\lambda_0^* = \alpha \times \lambda_0$, where $\text{IC}_{50}$ is the extracellular antibiotic concentration that leads to 50% growth inhibition, a common measure of drug sensitivity (S1 Appendix). Here, we call $\alpha$ the response parameter, as it describes the dose-response curve shape: The higher the value taken by $\alpha$, the shallower the dose-response curve (Fig 2C).

Since Eq (5) is cubic in the relative growth rate $y$, there are generally either one or three real solutions for $y$ (Fig 2C). This indicates that there is a parameter regime in which the dynamical system can exhibit bistability [10, 16]. Previous studies identified the bistable parameter regions numerically or in closed expression with many parameters [10, 14]. Notably, the rescaling shown above enables the exact calculation of the bifurcation point (see S1 Appendix): When $\alpha < \alpha_{\text{crit}} = 2/(3\sqrt{3}) \approx 0.385$ the system can be bistable (Fig 2D), *i.e.*, there is a region of concentrations with stable solutions at two different growth rates. At the border of this region, the growth rate sharply declines when a critical concentration is exceeded. It is difficult to measure such steep dose-response curves experimentally since very low growth rates are challenging to detect and quantify. Additionally, bistability cannot be observed in population-level experiments since the high-growth-rate branch will quickly dominate the population; single-cell experiments are needed to observe growth bistability [17]. On the other hand, if the antibiotic concentration can be varied during the experiment, bistability can be tested by determining the hysteresis of the response, as observed for synthetic gene networks [18].

Steep dose-response curves ($\alpha < \alpha_{\text{crit}}$) occur for antibiotics with tight binding to the ribosome ($K_D \to 0$) or inefficient efflux ($p_{\text{out}} \to 0$). Alternatively, if these two quantities are growth-rate invariant, dose-response curves become steeper with increasing growth rate in the absence of drug, as $\alpha \propto 1/\lambda_0$. For typical values of the relevant parameters (discussed above), $\alpha$ ranges from 0 to $\sim 10$. We have experimentally observed values of $\alpha$ for different translation inhibitors in the range $0 - 2$ [12]. In the limit $\alpha \gg 1$, Eq (5) simplifies into $y = 1/(1 + c)$; if $\alpha \to 0$ then this expression becomes $y = (1 + \sqrt{1 - c})/2$ for $c < 1$, [10]. This biophysical model for a single translation inhibitor provides the foundation for a predictive theory of multiple drug interactions between different translation inhibitors.

## Model for interaction between two translation inhibitors

**Dynamical system describing the binding of antibiotics and the physiological state of the cell.** When combinations of two different translation inhibitors are present, each ribosome can be bound by either of them alone or by both simultaneously. To generalize the model described in the previous section to this situation, we need to introduce additional populations of ribosomes. Extending the mathematical model [Eq (3)] to two translation

inhibitors yields:

$$\frac{\mathrm{d}a_i}{\mathrm{d}t} = -\lambda a_i + f_i(r_u, r_{b,i}, a_i) + \delta_{\mathrm{off},i}k_{\mathrm{off},i}r_d - \delta_{\mathrm{on},i}k_{\mathrm{on},i}a_i r_{b,\bar{i}} + J_i(a_{\mathrm{ex},i}, a_i),$$ (6a)

$$\frac{\mathrm{d}r_{b,i}}{\mathrm{d}t} = -\lambda r_{b,i} - f_i(r_u, r_{b,i}, a_i) + \delta_{\mathrm{off},\bar{i}}k_{\mathrm{off},\bar{i}}r_d - \delta_{\mathrm{on},\bar{i}}k_{\mathrm{on},\bar{i}}a_{\bar{i}} r_{b,i},$$ (6b)

$$\frac{\mathrm{d}r_d}{\mathrm{d}t} = -\lambda r_d + \sum_{i=A,B}\delta_{\mathrm{on},i}k_{\mathrm{on},i}a_i r_{b,\bar{i}} - \sum_{i=A,B}\delta_{\mathrm{off},i}k_{\mathrm{off},i}r_d$$ (6c)

$$\frac{\mathrm{d}r_u}{\mathrm{d}t} = -\lambda r_u + \sum_{i=A,B}f_i(r_u, r_{b,i}, a_i) + s(\lambda).$$ (6d)

The terms $f_i(r_u, r_{b,i}, a_i)$ and $J_i(a_{\mathrm{ex},i}, a_i)$ describe the first binding step and membrane transport of antibiotic $i$, respectively. The additional terms $\delta_{\mathrm{off},i}k_{\mathrm{off},i}r_d$ and $\delta_{\mathrm{on},i}k_{\mathrm{on},i}a_i r_{b,\bar{i}}$ describe the unbinding of antibiotic $i$ from double-bound ribosomes $r_d$ and the binding of antibiotic $i$ to ribosomes already bound by the other antibiotic $\bar{i}$ (*e.g.*, for antibiotics $A$ and $B$, $\bar{A} = B$), respectively. The dimensionless parameters $\delta_{\sigma,i}$ with $\sigma \in \{\mathrm{on, off}\}$ denote the relative change of the rate of forward and reverse binding of antibiotic $i$ to ribosomes already bound by the other antibiotic. When the binding kinetics for both antibiotics are independent, all $\delta_{\sigma,i} = 1$. When both antibiotics compete for the same binding site on the ribosome, $\delta_{\mathrm{on},i} = 0$. In general, the parameters $\delta_{\sigma,i}$ can vary continuously to capture any changes in ribosome binding of one antibiotic due to the binding of the second, as long as the binding is described by mass-action kinetics.

What is the main consequence of including the double-bound ribosomes? Below, we show that in the absence of double-bound ribosomes, drug interactions are generally expected to be additive. If we assume that no double-bound ribosomes can form, *e.g.*, by setting $\delta_{\mathrm{on},i} = 0$, Eq (6c) becomes equal to zero and all terms associated with the second binding event disappear. To show that this situation necessarily yields an additive drug interaction, we examine the system along an isobole. At fixed growth rate, *i.e.*, along an isobole, $r_u = \lambda/\kappa_t + r_{\mathrm{min}}$ is constant. This implies that the total concentration of ribosomes bound by either antibiotic (*i.e.*, $r_b = r_{b,A} + r_{b,B}$) remains constant for all different concentration pairs $(c_A, c_B)$ along the isobole. In steady state, the concentration of ribosomes bound by antibiotic $i$ is $r_{b,i} = a_i \times \xi_i$, where $\xi_i = k_{\mathrm{on},i}\lambda/[(k_{\mathrm{off},i} + \lambda)\kappa_t]$. The bound ribosome concentration reads

$$r_b = r_{b,A} + r_{b,B} = r_{\mathrm{tot}} - r_u = \Delta r(1 - \lambda/\lambda_0)$$ (7)

where we have taken into account Eqs (1) and (2). We express $a_i$ as a function of $a_{\mathrm{ex},i}$ from Eq (6a), which yields:

$$a_i = a_{\mathrm{ex},i}\frac{p_{\mathrm{in},i}}{\underbrace{[\lambda(k_{\mathrm{on},i}/\kappa_t + 1) + p_{\mathrm{out},i} - \xi_i k_{\mathrm{off},i}]}_{\Upsilon_i}}.$$ (8)

The proportionality constant $\Upsilon_i$ in this expression depends only on $\lambda$ and kinetic parameters; in particular, it is independent of the concentration of the other antibiotic. Since $r_b = r_{b,A} + r_{b,B} = \xi_A a_A + \xi_B a_B$ [Eq (7)], it follows that

$$\Delta r\left(1 - \frac{\lambda}{\lambda_0}\right) = a_{\mathrm{ex},A}\Upsilon_A\xi_A + a_{\mathrm{ex},B}\Upsilon_B\xi_B,$$ (9)

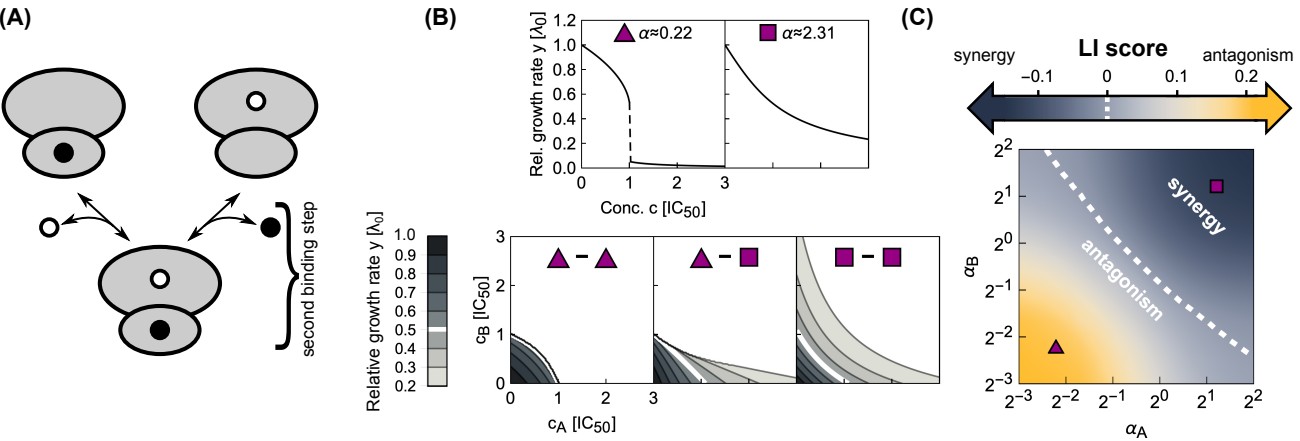

**Fig 3. Biophysical model of two antibiotics that can bind the ribosome simultaneously produces different types of drug interaction.** (A) Schematic: Ribosomes already bound by a single antibiotic (black and white circles) can be bound by another one. If the binding is independent of the presence of an already bound antibiotic, the second binding step follows the same kinetics as for a single antibiotic. (B) Examples of dose-response curves of different steepness and corresponding dose-response surfaces calculated from the model. Top: Dose-response curves with low or high $\alpha$ are steep (left) or shallow (right), respectively. Bottom: Depending on the shape of the dose-response curves of the antibiotics that are combined, the calculated drug interactions range from antagonism (left, low $\alpha$) to synergy (right, high $\alpha$). Combining antibiotics with different $\alpha$ results in a dose-response surface of more complicated shape (middle). (C) Phase diagram of drug interactions: *LI* score for dose-response surfaces of antibiotic pairs with response parameters $\alpha_A$, $\alpha_B$; white dashed line shows additive interactions (*LI* = 0). The left- and right-hand antibiotic pairs from (B) are shown by a purple triangle and a purple square, respectively.

which describes an isobole because the coefficients $\Upsilon_i \xi_i$ are independent of the other antibiotic. This argument shows that additivity generally occurs when double-bound ribosomes cannot form. Additionally, this confirms that the model correctly predicts additivity when the antibiotic is combined with itself–double-bound ribosomes cannot form in this case.

In the limit where $k_{\text{off}}, p_{\text{out}} \gg \lambda$, *i.e.*, $\alpha \to \infty$, Eq (9) becomes $c_A + c_B = \lambda_0/\lambda - 1$, where $c_i = a_{\text{ex},i}/\text{IC}_{50,i}$. To derive this expression, we noted the definitions of $\alpha$ and $\text{IC}^*_{50,i}$ used for a single antibiotic (see preceding section) as well as that $\text{IC}_{50,i} \approx \text{IC}^*_{50,i}\alpha/2$ for $\alpha \gg 1$ (S1 Appendix). This simplified expression clearly shows the linear dependency between external drug concentrations that result in growth rate $\lambda$.

To study the effect of double-bound ribosomes (Fig 3A), we systematically calculated dose-response surfaces for both competitive and independent binding (Fig 3B; S1 Appendix). The Loewe interaction score (*LI*) is a convenient way to characterize the type and strength of drug interactions by a single number, with negative values corresponding to synergy and positive values to antagonism [12]. The *LI* score quantifies the interaction using the volume under the dose-response surface:

$$LI = \log\left(\frac{\int y(c_A, c_B)\,\mathrm{d}c_A\mathrm{d}c_B}{\int y_{\text{add}}(c_A, c_B)\,\mathrm{d}c_A\mathrm{d}c_B}\right), \tag{10}$$

where $y_{\text{add}}$ is the response surface of the additive expectation, which is calculated directly from the responses to the individual drugs (see S1 Appendix). By calculating the *LI* score of the dose-response surfaces for varying response parameters $\alpha_A$, $\alpha_B$ of the two antibiotics that are combined, we determined the complete phase diagram of drug interactions (Fig 3C). This procedure revealed that antagonism generally occurs for combinations of antibiotics with steep dose-response curves, and the interaction becomes additive and then synergistic with increasing response parameters (Fig 3B). The phase diagram (Fig 3C) is a useful tool, because we can

obtain the null expectation for the drug interaction between antibiotics with known response parameters $\alpha$ simply by identifying their location in the phase diagram. The transition between both interaction regions is smooth, which implies a gradual transition of the drug interaction type. Because the *LI* score is an integrative measure of drug interaction strength, surfaces with $LI \approx 0$ are not required to be perfectively additive but rather need antagonistic and synergistic effects to cancel each other.

The fact that combinations of antibiotics that bind the ribosome irreversibly or are poorly pumped out of the cell yield antagonism can be understood intuitively. Consider a situation in which the antibiotics A and B are added to a bacterial population such that the concentration of antibiotic A far exceeds the concentration of antibiotic B. If at some point the majority of ribosomes is irreversibly bound by antibiotic A (due to its high intracellular concentration and/or irreversible binding), then antibiotic B is likely to bind to already inactivated ribosomes as well. Irreversibly-bound ribosomes thus effectively act as a "sponge" that soaks up antibiotics which can then no longer contribute to growth inhibition–a situation that results in antagonism.

What causes the transition from antagonism to synergy as $\alpha$ increases? Increasing $\alpha$ implies that the binding of the antibiotic becomes more and more reversible or efflux becomes high (*i.e.*, $K_D \, p_{\mathrm{out}} \to \infty$). If the growth rate is low due to inhibition, then $r_{\mathrm{tot}} \approx r_{\mathrm{max}}$ as ribosome synthesis is upregulated to its maximum. In this case, the typical rate of dilution by growth is much slower than that of antibiotic-ribosome binding and we obtain $a_i \approx a_{\mathrm{ex},i}\, p_{\mathrm{in},i}/p_{\mathrm{out},i}$. In this regime, we can derive an approximate solution that yields a synergistic dose-response surface, supporting the conclusion that qualitatively changing the binding kinetics alters the drug interaction type.

The system becomes linear and analytically solvable (see S1 Appendix). The growth rate is then:

$$
\begin{aligned}
y &\approx \frac{\lambda_{\mathrm{max}}/\lambda_0}{\{1 + c_A[(\alpha_A^2 + 1)\lambda_{\mathrm{max}}]/(\alpha_A^2 \lambda_0)\}\{1 + c_B[(\alpha_B^2 + 1)\lambda_{\mathrm{max}}]/(\alpha_B^2 \lambda_0)\}} = \\
&= \frac{\lambda_{\mathrm{max}}/\lambda_0}{(1 + c_A')(1 + c_B')},
\end{aligned}
\tag{11}
$$

where $c_i' = c_i \times [(\alpha_i^2 + 1)\lambda_{\mathrm{max}}]/(\alpha_i^2 \lambda_0)$. This expression for $y$ is simply a product of relative responses, which is equivalent to the definition of Bliss independence but corresponds to a synergistic interaction according to Loewe definition (Fig 4). This approximate solution agrees well with the full numerical solution at lower growth rates and for antibiotics with higher $\alpha$ (Fig 4). Eq (11) becomes even simpler in the limit $\lambda_0 = \lambda_{\mathrm{max}}$ and $\alpha \to \infty$ as these two limits yield a product of two Langmuir-like equations with only relative concentrations $c_A$, $c_B$ as arguments, *i.e.*, $y = 1/[(1 + c_A)(1 + c_B)]$, which is independent of $\lambda_0$ and $\alpha$.

The results above can be experimentally verified. To predict the entire dose-response surface, we require only the response parameters $\alpha$, which can be obtained by fitting Eq (5) to individual dose-response curves. We recently performed this comparison and found good agreement of theory and experiment for most, but not all, drug pairs [12].

**Symmetric direct interactions on the ribosome amplify drug interactions.** We next asked how more general binding schemes, in which two different antibiotics can directly interact on the ribosome to stabilize or destabilize their binding affect the resulting drug interactions. Two antibiotics do not need to come into direct, physical contact to affect each other's binding: Allosteric effects (*i.e.*, changes in ribosome structure due to antibiotic binding) can produce the same result. In the most plausible scenario, the antibiotics affect each other's binding in a symmetric way, *i.e.*, $\delta_{\sigma,i} = \delta_{\sigma,\bar{i}}$ (Fig 5A). For example, the antibiotics lankamycin and

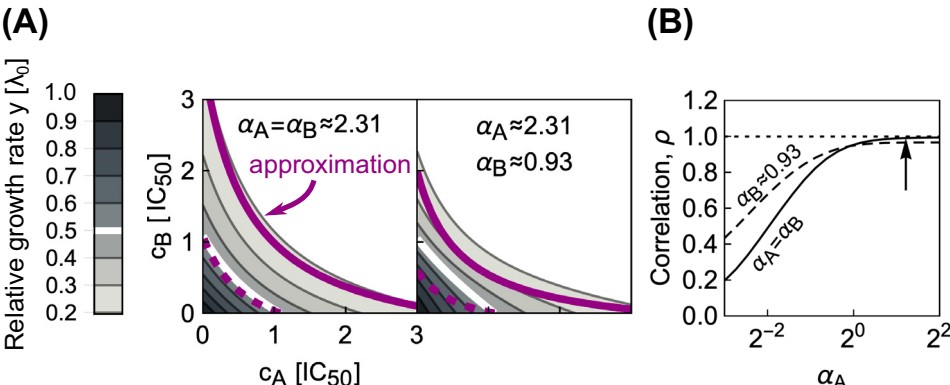

**Fig 4. Combining antibiotics with shallow dose-response curves yields synergistic drug interactions.** (A) Comparisons of numerically calculated dose-response surfaces and approximate solution. Purple isoboles (dashed and solid lines correspond to 50% and 20% relative growth rate, respectively) show the approximate solution on top of the dose-response surface calculated from the biophysical model (gray scale). Examples are shown for two pairs of antibiotics with identical (left) or different $\alpha$ (right). (B) Pearson correlation $\rho$ between approximate and numerically calculated growth rate, evaluated for $121 \times 121$ equidistant concentration pairs. Solid and dashed line correspond to the cases with identical or different $\alpha$, respectively. The correlation increases for antibiotics with higher $\alpha$. The arrow shows $\alpha$ for the example on the left in (A).

lankacidin (which interact synergistically [19]) are near each other when bound to the ribosome; their binding is stabilized by a direct physical interaction [20, 21]. Stabilization of binding could also increase the sequestration of tightly binding antibiotics or "lock" an antibiotic that would rapidly unbind on its own in the bound state, thus potentially promoting prolonged inhibition of the ribosome. In contrast, the two antibiotics may also mutually destabilize their binding to the ribosome. The limiting case of this scenario is competition for the same binding

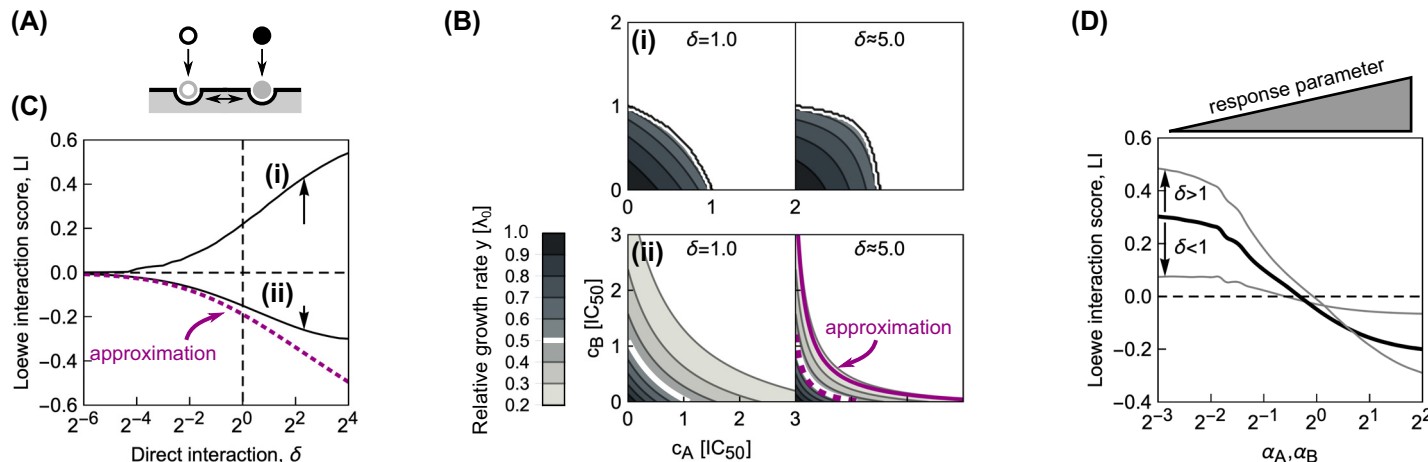

**Fig 5. Direct interactions between antibiotics on the ribosome can amplify drug interactions.** (A) Schematic of antibiotics symmetrically affecting their binding on the ribosome. (B) Changes in the shape of the dose-response surfaces for pairs of antibiotics with (i) identical $\alpha = 2^{-5}$ and (ii) identical $\alpha = 2^2$, when $\delta$ is increased from 1 (left) to $\approx 5.0$ (right). Purple dashed and solid line in the bottom-right panel show the approximate solution in Eq (14) for the 50% and 20% isoboles, respectively. (C) Increase in absolute value of the *LI* score as a function of $\delta$ for pairs of antibiotics with different response parameters. Solid lines (i) and (ii) correspond to the examples in (B); arrows show increase in |*LI*| at $\delta \approx 5.0$. Note, that for both antibiotic combinations, *LI* collapses to 0 for competitive binding, *i.e.*, $\delta = 0$. The dotted line shows the *LI* score calculated using the approximate solution in Eq (14). (D) Diagonal cross-section ($\alpha_A = \alpha_B$) through the phase diagram for different $\delta$. Black solid line corresponds to the case of independent binding (*cf.* Fig 3C); the two gray lines show examples with either $\delta < 1$ or $\delta > 1$. Irrespective of drug interaction type, the drug interaction is amplified for $\delta > 1$ and weakened for $\delta < 1$.

site. To investigate such effects systematically, we computed dose-response surfaces for antibiotics with different response parameters $\alpha$ and varying kinetics for the second binding step.

We focused on pairs of antibiotics in which both drugs either have low or high $\alpha$, corresponding to steep or shallow dose-response curves, respectively. Numerical solutions for continuously varying $\delta_{\mathrm{on},i} = \delta$ at fixed $\delta_{\mathrm{off},i} = 1$ (Fig 5B and 5C) showed that a stabilizing interaction ($\delta > 1$) enhances the resulting drug interaction. If the drug interaction is antagonistic for $\delta = 1$, stabilization amplifies this antagonism; synergistic interactions are amplified analogously (Fig 5B). If one antibiotic destabilizes the binding of the other, *i.e.*, $\delta < 1$, a smooth transition to additivity occurs, independent of whether the dose-response curve of the antibiotic pair is steep or shallow (Fig 5C). This result is further corroborated by fixing $\delta$ and continuously varying $\alpha$ for the combined antibiotics (Fig 5D). Taken together, these numerical results indicate that direct positive interactions of translation inhibitors on the ribosome ($\delta > 1$) amplify the drug interaction that occurs in the absence of such direct interactions, irrespective of drug interaction type.

Experimentally, these results suggest that the presence of direct interactions on the ribosome between bound antibiotics would manifest in stronger synergy than predicted for independent binding if the response parameters of the combined antibiotics are sufficiently high (S1 Appendix). However, for a direct prediction of the dose-response surface one requires the value for the parameter $\delta$, which can only be extracted from dedicated biochemistry measurements.

To corroborate these numerical results, we investigated the limit of reversibly binding antibiotics with rapid binding kinetics at low growth rates as for Eq (11). In this limit, there is again an analytical solution for the dose-response surface:

$$y \approx \frac{\lambda_{\max}/\lambda_0 (\delta_{\mathrm{off},A} + \delta_{\mathrm{off},B} + c'_A \delta_{\mathrm{on},A} \delta_{\mathrm{off},B} + c'_B \delta_{\mathrm{on},B} \delta_{\mathrm{off},A})}{(1 + c'_A + c'_B)[\delta_{\mathrm{off},B} \Phi_A + \delta_{\mathrm{off},A} \Phi_B] + c'_A c'_B \Phi_{AB}}, \tag{12}$$

where $\Phi_{AB} = [\delta_{\mathrm{on},A} + \delta_{\mathrm{on},B} + \delta_{\mathrm{on},A} \delta_{\mathrm{on},B} (c'_A + c'_B)]$, $\Phi_B = (1 + c'_B \delta_{\mathrm{on},B})$, and $\Phi_A = (1 + c'_A \delta_{\mathrm{on},A})$. This closed-form expression facilitates the analysis of several limiting cases. For example, if the antibiotics mutually stabilize their binding to the extreme extent that they cannot detach from the double-bound ribosomes anymore ($\delta_{\mathrm{off},i} = 0$), Eq (12) returns $y = 0$ indicating strong synergism. In contrast, prohibiting the formation of double-bound ribosomes by setting $\delta_{\mathrm{on},i} = 0$, yields

$$y_{\mathrm{add}} \approx \frac{\lambda_{\max}/\lambda_0}{1 + c'_A + c'_B}, \tag{13}$$

which corresponds to Loewe additivity and is different from Bliss independence obtained in Eq (11). This corroborates the previous result that competitively binding antibiotics interact additively. For the case $\delta_{\mathrm{on},i} = \delta$ and $\delta_{\mathrm{off},i} = 1$ the expression in Eq (12) simplifies to:

$$y \approx \frac{\lambda_{\max}/\lambda_0}{1 + c'_A + c'_B + \delta c'_A c'_B}, \tag{14}$$

which becomes Eq (11) if $\delta = 1$. We can further show that the effect of increasing $\delta$ on drug interaction strength depends on the concavity of the individual dose-response curves: for response parameters $\alpha > 2$, increasing $\delta$ amplifies synergy (see S1 Appendix). Overall, this analysis corroborates the general result that direct stabilizing interactions of reversibly-binding antibiotics on the ribosome amplify synergistic interactions, while destabilizing interactions weaken them up to the point where any drug interaction becomes additive.

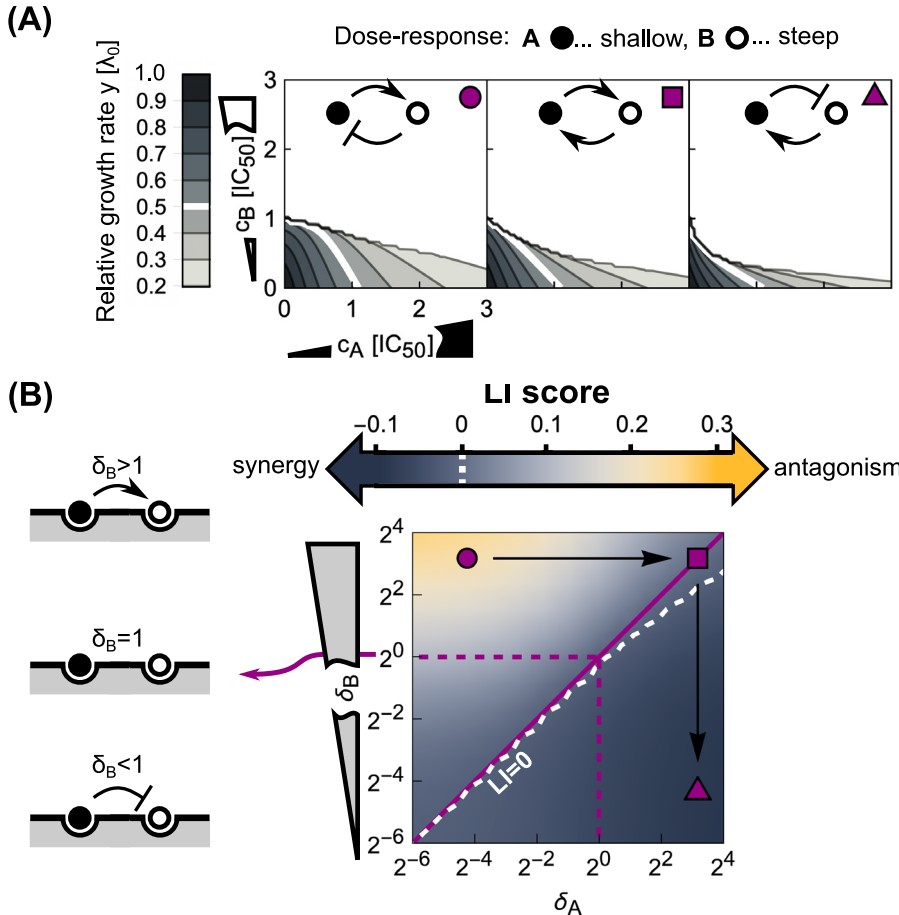

**Fig 6. Asymmetric direct interactions reshape the phase diagram of drug interactions.** (A) Dose-response surfaces for different instances of asymmetric direct interaction and response parameters; insets on top show schematics of the type of direct interaction and top-right symbols correspond to those in (B). Antibiotics with shallow ($\alpha_A = 2^2$) and steep ($\alpha_B = 2^{-3}$) dose-response curves are shown by black and white disks, respectively. Left: Antagonism occurs when an antibiotic with a steep dose-response asymmetrically hinders the binding of another one with a shallow dose-response, which in turn promotes the binding of the former. Middle: Symmetrizing the direct interaction almost completely abolishes antagonism. Right: Inverting the scenario from the left-most panel results in mild synergy. (B) Phase diagram of drug interactions for asymmetric direct interactions between antibiotics with different response parameters ($\alpha_A = 2^2$ and $\alpha_B = 2^{-3}$) profoundly affect the resulting drug interaction: A continuous transition from antagonism to synergy occurs (white dashed line denotes $LI = 0$). Purple symbols show the examples from (A) in the phase diagram.

**Asymmetric direct interactions alter the phase diagram.**   More generally, direct interactions between the antibiotics on the ribosome could be asymmetric. For example, binding of only one of the antibiotics could trap the ribosome in a conformation that facilitates the binding of the other antibiotic but not *vice versa*. To investigate such effects, we fixed $\delta_{\text{off},i} = 1$ and varied $\delta_{\text{on},i}$ for antibiotics with different response parameters $\alpha$ (Fig 6). The resulting difference in kinetic parameters describes an asymmetric direct interaction on the ribosome. We systematically calculated the shape of the dose-response surface for this situation.

When antibiotics with identical response parameters $\alpha$ are combined, the same trend as for symmetric direct interactions occurs: Increasing $\delta_{\text{on}}$ enhances the drug interaction. For combinations of antibiotics with different dose-response curve shapes, asymmetric direct interactions on the ribosome result in a different behavior (Fig 6A and 6B). If an antibiotic with a

steep dose-response curve asymmetrically hinders the binding of an antibiotic with a shallow dose-response, while the binding of the former is stabilized by the latter, antagonism emerges (Fig 6A). In contrast, synergy occurs if the roles of the antibiotics are inverted (Fig 6A). The latter can be rationalized by interpreting the direct interaction on the ribosome as a change of the effective binding characteristics of the antibiotics. Specifically, in the case where the steep-response antibiotic promotes the binding of the shallow-response antibiotic, the latter will in turn destabilize the binding of the former–effectively, the steep-response antibiotic will thus behave as if it had a shallower response. As a result, synergy occurs–exactly as expected when two shallow-response antibiotics are combined (Fig 3C). In the opposite situation, the binding of the shallow-response antibiotic becomes even looser and the binding of the steep-response antibiotic is stabilized. From the phase diagram in Fig 3C, antagonism is the expected outcome in this case as we combine antibiotics with steep and shallow responses, respectively. Taken together, these results show how complicated direct interactions between antibiotics bound to their target can lead to unexpected emergent drug interactions.

**Relation to mechanism-independent models of higher-order drug interactions.** The biophysical model described above can predict the pairwise drug interactions that are needed to apply recently proposed mechanism-independent models for higher-order drug interactions [8]. While a detailed analysis of higher-order drug interactions is beyond the scope of this article, it is instructive to demonstrate how the pairwise interactions bridge the gap between responses to individual drugs and higher-order drug combinations. In the framework of Ref. [8], higher-order drug interactions can be predicted using an entropy maximization method, in which the joint drug effects are fully determined by the responses to the individual drugs ($y_i$) and their pairwise combinations ($y_{ij}$):

$$y_{ABC} = y_A y_{BC} + y_B y_{AC} + y_C y_{AB} - 2y_A y_B y_C. \tag{15}$$

In the limit of slow growth used previously, it is straightforward to analyze the effects of higher-order drug combinations. The approximate results below are based on the assumptions that: (i) the growth rate is directly proportional to the concentration of unblocked ribosomes, (ii) growth rate is nearly zero, (iii) the intracellular drug concentration depends only on transport kinetics (*i.e.*, $a \approx a_{ex} p_{in}/p_{out}$), and (iv) the growth rate is directly proportional to the concentration of unblocked ribosomes (Fig 7A). Under these assumptions, analytical solutions can be obtained. For example, we can construct a system of differential equations describing the binding of three different antibiotics ($A$, $B$, $C$); the steady-state solution of this system is:

$$y \approx \frac{\lambda_{max}/\lambda_0}{(1 + c_A')(1 + c_B')(1 + c_C')}. \tag{16}$$

To derive this result, we considered three different kinds of single- and double-bound ribosomes as well as triple-bound ribosomes (Fig 7B); for simplicity, all binding steps were considered to be independent of already bound antibiotics. To verify the consistency of the mechanism-independent model with this fully specified but approximative mechanistic model, we need to obtain responses to individual drugs $y_i$ and pairwise combinations $y_{ij}$. We obtain responses $y_i = 1/(1 + c_i')$ and $y_{ij} = 1/[(1 + c_i')(1 + c_j')]$ from Eq (16) by setting all $c_j'$ with ($j \neq i$) and all $c_k'$ with ($k \neq i, j$) to zero, respectively. If the mechanism-independent expression is consistent with our simplified model, then plugging these responses into Eq (15) should yield Eq (16), which is indeed the case.

Next, we tested if the mechanism-independent formula for three drugs [Eq (15)] can be reconciled with our model when there is one direct competitive interaction on the ribosome. If the antibiotics $B$ and $C$ cannot bind to the ribosome simultaneously, the solution of the

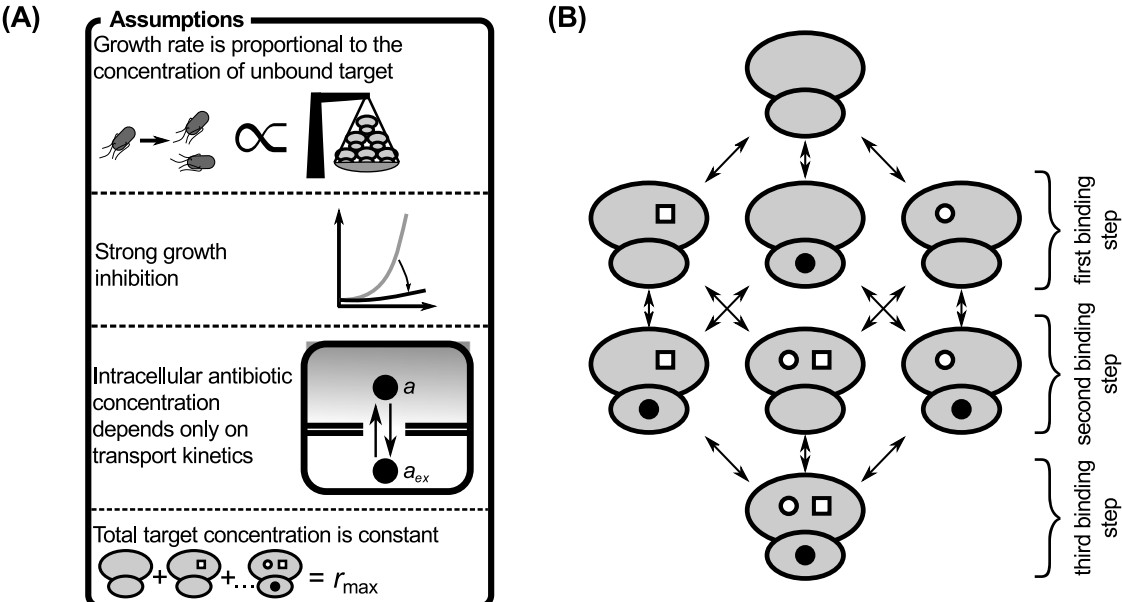

**Fig 7. Assumptions and binding kinetics diagram underlying the calculation of higher-order drug interactions.**
(A) Assumptions that simplify the system to allow obtaining a closed solution. (B) Binding kinetics diagram shows allowed transitions between ribosome subpopulations. Different symbols on the ribosomes denote different antibiotics.

approximate system is:

$$y_{ABC} = \frac{\lambda_{\max}/\lambda_0}{(1 + c'_A)(1 + c'_B + c'_C)}. \tag{17}$$

By following same reasoning as for the independent case above, we see that the mechanism-independent and the simplified mechanistic model are consistent.

The assumptions described above can be generalized to antibiotics with other modes of action, provided that the combined antibiotics bind to the same target. For example, if growth is limited by a specific enzyme due to antibiotic inhibition, we can consider the growth rate to be proportional to the abundance of this limiting enzyme. If the enzyme concentration does not change, the approximative mathematical framework from this section is applicable to antibiotics targeting this enzyme. These results provide a potential mechanistic explanation for the apparent validity of the mechanism-independent model, at least for combinations of antibiotics binding the same target.

## Extensions of the model

Other phenomena than those treated so far can shape drug interactions. Below, we discuss two cases in which (i) antibiotics perturb translation in orthogonal ways and (ii) the expression of antibiotic resistance genes alters a drug interaction. While certainly not exhaustive, these two cases illustrate relevant extensions of the model.

**Effects of antibiotic-induced starvation.** Translation inhibitors target the protein synthesis machinery, which is carefully regulated in response to changes in the nutrient environment [22]. Thus, if an antibiotic effectively interferes with cellular state variables that represent the nutrient environment, it should be possible to predict its effect on the action of a translation inhibitor and, in turn, its drug interaction with a translation inhibitor.

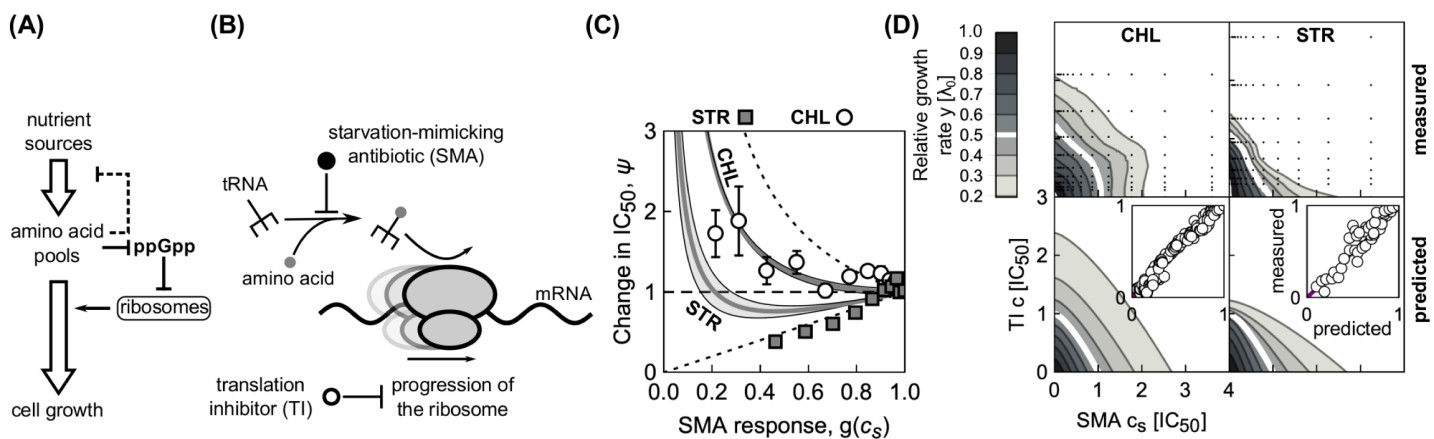

**Fig 8. Effects of a starvation-mimicking antibiotic on the efficacy of translation inhibitors.** (A) Schematic: Simplified regulation of translation coordination. Nutrients are transported into the cell, where they serve as a source of amino acids. These amino acids are required for tRNA charging. Oversupply of amino acids leads to down-regulation of the nutrient transport and processing machinery, and depletion of the intracellular signaling molecule ppGpp (guanosine tetraphosphate). This in turn de-represses the expression of the translation machinery, which increases the overall translation capacity, leading to faster growth. In contrast, if amino acids are in short supply, the translation machinery is down-regulated. (B) Translation inhibitors (TI) inhibit progression of the ribosome, while a starvation-mimicking antibiotic (SMA) perturbs the amino acid supply. The ribosome progresses along the mRNA (black wavy line), if charged tRNAs (black fork with gray circle) deliver amino acids (gray circles) at a sufficient rate to support the rapid synthesis. A starvation-mimicking antibiotic inhibits tRNA charging and thus mimics amino acid depletion, a hallmark of starvation. (C) Dependence of relative change in IC$_{50}$ on SMA inhibition ($\psi$ = IC$_{50}$/IC$_{50,F}$). Example solutions of Eq (18) were calculated for chloramphenicol (CHL; top solid line with $\alpha_F$ = 1.04, white circles show experimental data) and streptomycin (STR; bottom solid line with $\alpha_F$ = 0.46, gray squares show experimental data). The gray areas correspond to the confidence intervals as obtained from standard errors in $\alpha$. Values for response parameters $\alpha$ and their standard errors are from Ref. [12]. In the experiments, mupirocin (MUP) was used as SMA. The horizontal dashed line indicates no change with respect to $g(c_s)$. Error bars correspond to standard errors in IC$_{50}$ as obtained from fitting; where error bars are not visible, they are smaller than the symbol size. (D) Measured (top) and predicted (bottom) dose-response surfaces for CHL-MUP (left) and STR-MUP (right). Insets show scatter-plots of predicted and measured non-zero growth rates.

Bacterial growth strongly depends on the availability and quality of nutrients. Protein synthesis requires that amino acids are delivered to the translation machinery (ribosomes) by dedicated proteins [elongation factors (EF-Tu)] [23]. The latter bring charged tRNAs (*i.e.*, tRNAs with an attached amino acid) to the ribosome (Fig 8B). tRNAs are charged (*i.e.*, amino acids are attached to them) by tRNA synthetases. Usually, the supply and demand of amino acids can be considered to be nearly optimally regulated [22] (Fig 8A). However, under starvation, a mismatch between the supply and demand of amino acids occurs [24]. Bacteria respond to amino acid starvation by triggering the stringent response. This starvation response is primarily controlled by the alarmone ppGpp (guanosine tetraphosphate) which down-regulates the expression of the translation machinery (Fig 8A) [25]. Amino acid starvation is reflected in reduced tRNA charging and usually occurs when the nutrient environment becomes poor. However, amino acid starvation can also be caused by a *starvation-mimicking antibiotic* (SMA) that blocks tRNA synthetases (Fig 8B) [26, 27].

We can capture the effect of an SMA in our model and thus make predictions for the drug interactions between an SMA and translation inhibitors. To this end, we assume that the growth rate in the absence of drug $\lambda_0$, which characterizes the quality of the nutrient environment in Eq (4), depends on the concentration $c_s$ of the SMA only. Under this assumption, the growth rate in the simultaneous presence of an SMA and a translation inhibitor can be derived directly from the previous results for a single antibiotic [Eq (5)]. In the absence of translation inhibitor, the growth rate is given by the dose-response function of the SMA $g(c_s)$. Since IC$_{50}$ $\propto (\alpha^2 + 1)/\alpha$ and $\alpha = \alpha_F/g(c_s)$, the relative change in IC$_{50}$ at constant SMA concentration becomes

$$\psi = \frac{\text{IC}_{50}}{\text{IC}_{50,F}} = \frac{\alpha_F^2 + g^2(c_s)}{(\alpha_F^2 + 1)g(c_s)}, \tag{18}$$

where $\alpha_F$ and $\text{IC}_{50,F}$ are $\alpha$ and $\text{IC}_{50}$ in the absence of the SMA, respectively. It follows that $\psi$ increases monotonically with SMA inhibition if $\alpha_F > 1$; this condition is obtained by solving $\partial_g \psi = 0$ for $g(c_s) \leq 1$. If $\alpha_F \leq 1$, then the minimal $\psi$ is reached at $g(c_s) = \alpha_F$. We further note that two functional limits exist: in the limits $\alpha \to 0$ and $\alpha \to \infty$, Eq (18) becomes $\psi = g(c_s)$ and $\psi = 1/g(c_s)$, respectively.

The dose-response curve for a single antibiotic is given by $y = f(\alpha, c)$ [a solution of Eq (5)]. Since we know how $\text{IC}_{50}$ [Eq (18)] and $\alpha$ change as a function of $g(c_s)$, we can evaluate the entire dose-response surface:

$$y(c, c_s) = g(c_s) \times f(\alpha_F / g(c_s), c/\psi). \tag{19}$$

Eq (19) deviates from a simple multiplicative expectation since the SMA affects the parameters of the dose-response curve of the translation inhibitor as well. This result illustrates how deviations from Bliss independence occur when the combined drugs mutually affect their dose-response characteristics. Hence, this generalization of our model makes non-trivial quantitative predictions for drug interactions that occur between an SMA and translation inhibitors.

A specific example of an SMA is the antibiotic mupirocin (MUP), which reversibly binds to isoleucin tRNA synthetase and prevents tRNA charging [28]. MUP, which is used against clinically problematic methicillin-resistant *Staphylococcus aureus* (MRSA) infections [29], induces the stringent response [26, 27] and can thus be used to test our theoretical prediction. To this end, we measured the change of the $\text{IC}_{50}$ of *E. coli* at different levels of growth inhibition caused by MUP $g(c_s)$ for two translation inhibitors: chloramphenicol (CHL) and streptomycin (STR); see S1 Appendix. CHL and STR have different response parameters $\alpha$ [12], which leads to different dependencies of $\psi$ on SMA inhibition (Fig 8C), an effect that is closely related to the results for different nutrient environments in Ref. [10]. We note, that while the model predicts a shallower decrease in $\psi$ for STR compared to the observed one, the data lies very close to the border of the limiting case. This suggests that the discrepancy originates predominantly from a poor fit of $\alpha$; fitting Eq (5) to steep dose-response curves is notoriously difficult due to the abrupt drop and the scarce data at low growth rate. Similarly, we measured the complete dose-response surfaces for both drug pairs. The theoretically predicted dose-response surfaces qualitatively agree with the experimentally observed ones (Fig 8D). In S1 Appendix we discuss further examples of pairwise antibiotic combinations in which a translation inhibitor is combined with a drug that alters the growth law parameters. Together, these results illustrate how our theoretical model can be extended to predict the effects of drug combinations beyond antibiotics that directly target the ribosome.

**Effect of constitutively expressed resistance genes.** Our results show that the steepness of the dose-response curve and the coupling between growth laws and antibiotic response play a key role in determining drug interactions. Dose-response curve steepness can change if genes that convey antibiotic resistance are present [17]. Thus, we investigated how the presence of such resistance genes affects the resulting drug interaction.

Bacterial resistance genes often code for dedicated enzymes that degrade the antibiotic or pump it out of the cell. Resistance genes can be constitutively expressed, *i.e.*, they lack specific regulation and their expression depends only on the state of the gene expression machinery. The expression of such constitutively expressed resistance genes (CERGs) under translation inhibition is quantitatively predicted by a theory based on bacterial growth laws [9, 17]. When the growth rate is varied by nutrient quality, the expression of a constitutively expressed gene $q$ linearly decreases until reaching zero at $\lambda_{max} = \Delta r \kappa_t$, *i.e.*, $q \propto (1 - \lambda_0/\lambda_{max})$ (Fig 9A). Yet, if the growth rate changes due to translation perturbation, expression decreases with decreasing

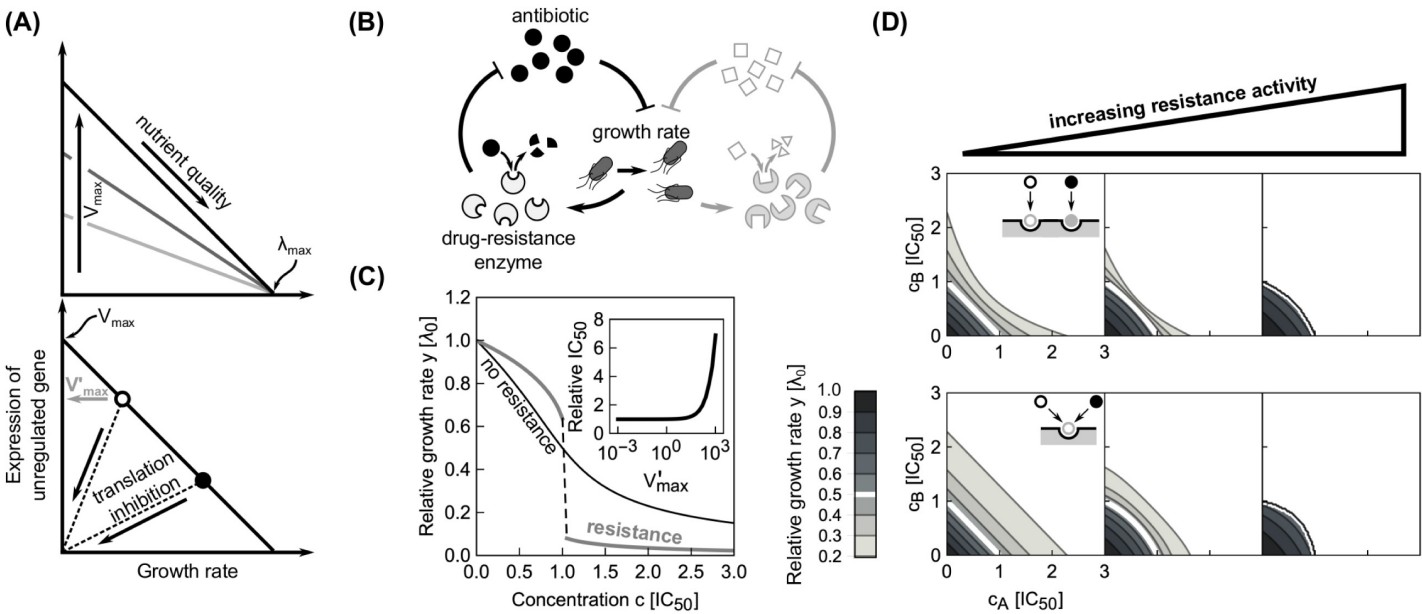

**Fig 9. Effects of constitutive resistance genes on the shape of dose-response curves and surfaces.** (A) Expression of unregulated (constitutive) gene depends on nutrient quality and degree of translation inhibition. Top: When growth is varied by nutrient quality, the expression of the gene decreases with increasing growth. The highest expression achieved in the limit of low growth rates is determined by $V_{max}$ (different shades of gray). When $\lambda_0 = \lambda_{max}$, expression ceases, invariantly of $V_{max}$. Bottom: In a fixed nutrient environment (circles), expression decreases as the growth rate decreases upon translation inhibition. The expression in the absence of antibiotic is $V'_{max} = V_{max}(1 - \lambda_0/\lambda_{max})$ [gray arrow; for the environment with lower $\lambda_0$ (white disk)]. (B) Schematic of positive feedback loop for unregulated antibiotic resistance gene. A drug-resistance enzyme degrades the antibiotic, thus reducing growth inhibition and boosting its own expression. However, if the antibiotic concentration exceeds the capacity of removal by the enzyme, growth rate starts to drop and so does the expression of the resistance enzyme, amplifying the growth rate drop. The lightly drawn part (right) illustrates how two antibiotics can get coupled via the growth-rate dependent loop. (C) Examples of dose-response curves in the presence or in the absence of a constitutively expressed resistance gene (CERG). Black line shows dose-response curve for $\alpha = 1$. When a CERG is added [$V'_{max} = 1000$ $\mu$M h$^{-1}$, $K_{rem} = 0.1$ $\mu$M the dose-response curve becomes steeper and exhibits an abrupt drop. Inset shows the increase in antibiotic concentration required to halve the growth rate relative to the no-CERG case as a function of $V'_{max}$. (D) Dose-response surface for independently (top) and competitively (bottom) binding antibiotics with $\alpha$ = 1; resistance activity $V'_{max}$ (assumed to be identical for both antibiotics) increases from left to right: 0, 100 and 950 $\mu$M h$^{-1}$. Concentration axes were rescaled with respect to the increased IC$_{50}$. Note the qualitative change in dose-response surface shape.

growth rate, *i.e.*, $q \propto \lambda/\lambda_0$ (Fig 9A). An experimentally verified mathematical model that is based on this growth rate-dependent expression predicts growth bistability (*i.e.*, coexistence of growing and non- or slowly-growing cells) in bacterial populations that constitutively express resistance genes [17]. In this model, the flux of antibiotic removal due to the resistance enzyme is described by

$$
\begin{aligned}
j_{rem} &= V_{max} \underbrace{\left(1 - \frac{\lambda_0}{\lambda_{max}}\right)}_{nutrient-dependent} \left(\frac{\lambda}{\lambda_0}\right) \times \frac{a}{a + K_{rem}}, \\
&= V'_{max}\left(\frac{\lambda}{\lambda_0}\right) \times \frac{a}{a + K_{rem}}
\end{aligned}
\tag{20}
$$

where $K_{rem}$ is a Michaelis-Menten dissociation constant and $V_{max}$ is the maximal antibiotic removal rate, which bundles the maximal enzyme abundance and catalytic rate per enzyme. In Eq (20), $V'_{max} = V_{max}(1 - \lambda_0/\lambda_{max})$.

Due to the linear relation between growth rate and the expression of the resistance gene, the rate of antibiotic removal decreases with decreasing growth rate under translation inhibition. This constitutes a positive feedback loop that leads to growth bistability (Fig 9B and 9C),

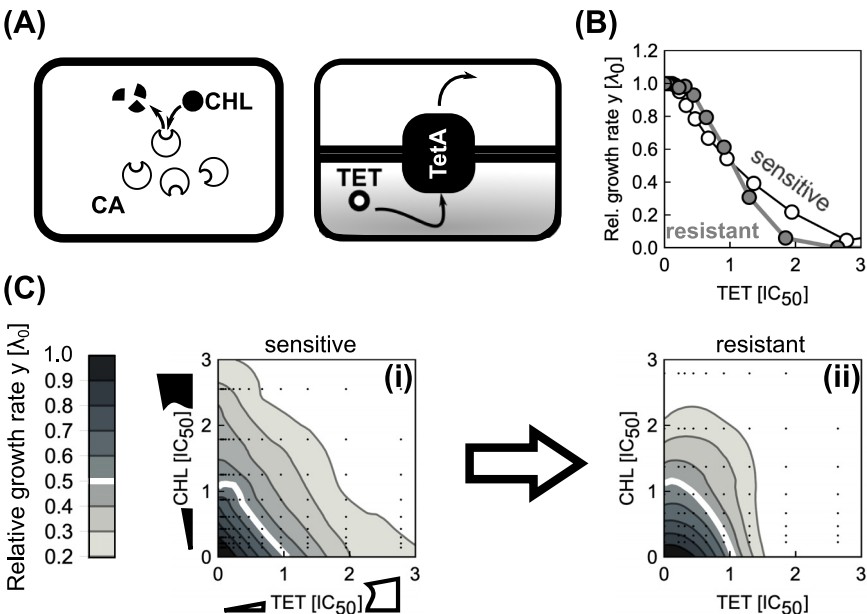

**Fig 10. Constitutively expressed resistance genes alter a drug interaction as predicted by theory.** (A) Schematic showing two common resistance mechanisms: Resistance can result from degradation of the drug [left: chloramphenicol acetyltransferase (CAT) degrades chloramphenicol (CHL)] or from drug efflux [right: an antibiotic efflux pump (TetA) removes tetracycline (TET) from the cell]. (B) Change in dose-response curve shape due to a constitutively expressed resistance gene. CHL dose-response curves of sensitive (white circles) and resistant strain (gray circles). (C) Measured CHL-TET dose-response surfaces for (i) sensitive and (ii) resistant strain. Concentrations were normalized to the $IC_{50}$ of respective strains. The strain with CERGs is 50.5 and 91.5 times more resistant to TET and CHL, respectively, as measured by increase of $IC_{50}$. Drug interaction changes from additive to antagonistic as suggested by theory (Fig 9D).

which is reflected in a steep dose-response curve of bacterial batch cultures [17]. However, note that for very high values of $K_{rem} \gg a$, Eq (20) becomes linear and the steepness of the dose-response curve decreases, rendering the otherwise bistable system monostable (see S1 Appendix).

By extending this scenario to a pair of antibiotics, we can directly test how the presence of resistance genes affects drug interactions. In the most relevant case, there are two CERGs each of which specifically provides resistance to one of the antibiotics. For simplicity, we assume that there is no cross-resistance, *i.e.*, each enzyme specifically degrades only one of the drugs (Fig 9B). We found that the synergistic interaction between two independently binding antibiotics with shallow dose-response curves turns slightly antagonistic due to the presence of resistance genes (Fig 9D, top). For competitively binding antibiotics, this effect becomes more pronounced (Fig 9D, bottom). In brief, our model predicts qualitative changes in drug interaction type when resistance genes are present.

To test this prediction, we constructed an *E. coli* strain (see S1 Appendix) that carries two constitutively expressed resistance genes. We chose TetA [a tetracycline (TET) efflux pump] and CAT [an enzyme that degrades chloramphenicol (CHL)], which were previously characterized in the context of bacterial growth laws (Fig 10A; [17]). Furthermore, the interaction between CHL and TET is additive. Our model predicts this interaction to change into antagonism when CERGs are present. Consistent with previous results [17], the steepness of the dose-response curve increased upon inclusion of each CERG (Fig 10B). We measured the dose-response surface of the sensitive and the double-resistant strain: Notably, the resistant

strain showed a clear antagonistic drug interaction, while this interaction was additive in the strain without CERGs (Fig 10C). This change to antagonism qualitatively agrees with the theoretical prediction (Fig 9D). This example shows how resistance genes can drastically alter drug interactions–a phenomenon caused by a non-trivial interplay of gene-expression and cell physiology predicted by our biophysical model.

In future work, this framework could be expanded to include resistance mechanisms other than the efflux and degradation of the drug. Other resistance mechanisms include target modification, overproduction of a target mimic (decoy), and factor-associated protection [20]. These mechanisms offer attractive modeling and experimental opportunities. On the modeling side, these additional mechanisms require the introduction of new sub-populations of ribosomes (modified or factor-associated) or target mimics. Experimentally, handling these highly resistant strains is challenging as minimal inhibitory concentrations approach the solubility limits of the relevant antibiotics, which requires fine-tuning of the expression system. Further, the effects we predict above should be growth environment-dependent, which follows from Eq (20): at low $\lambda_0$, the expression of CERG should increase and thus its effects should manifest. If $\lambda_0 \to \lambda_{max}$, then the effect of CERG should be less prominent and the drug interaction should resemble the WT one. While these extensions are of high basic and clinical importance, they are outside of the scope of this study. Here we included only the best-characterized examples that required minimal genetic intervention in the system.

## Discussion

We constructed a minimal biophysical model of antibiotic interactions that takes into account the laws of bacterial cell physiology. Most parameters in our model are constrained by established results or by the dose-response curves of the individual antibiotics that are combined (Fig 3). Our approach offers a scalable theoretical framework for predicting drug interactions: The number of parameters required for the independent binding model scales linearly with the number of antibiotics. This framework is readily generalized to combinations of more than two antibiotics. Ribosomal growth laws [9] were essential for building this predictive framework, highlighting the importance of quantitative phenomenological descriptions of physiological responses to drugs and other perturbations (Fig 2). The discovery of similar quantitative relations between physiological parameters and growth rate for other classes of antibiotics and other types of cells would greatly facilitate more general predictions of drug interactions.

Our work highlights the advantages of a physiologically relevant "null model," which captures all effects that are generally relevant for ribosome-binding antibiotics without trying to describe any molecular details of specific antibiotics (Fig 3). While general multiplicative (Bliss) or additive (Loewe) expectations are simple to construct, our work demonstrates that their utility as a reference has clear limitations. Specifically, our model shows that both are expected to be valid only in certain limits (Figs 4 and 5). Moreover, these standard null models do not capture known effects of antibiotic binding and growth physiology, which suffice to produce strong deviations from the standard null models. Our biophysical model captures these effects and thus offers an improved expectation for drug interactions. Generalizing this model to three drugs demonstrated that mechanism-independent predictions of higher-order interactions [8] are consistent with simplified first-order kinetics. In summary, our model serves as a bridge between mechanism-independent general predictions of drug interactions and elusive quantitative descriptions of detailed molecular mechanisms that capture the idiosyncrasies of each drug.

We showed that direct physical (or allosteric) interactions of antibiotics on their target do not necessarily lead to synergy (Fig 5 and S1 Appendix). Synergy only occurs if the dose-response curves of the individual drugs are sufficiently shallow. While this insight is not easily applied in the design of drug combinations, the identification of cooperatively binding drug pairs still has considerable potential. Our results highlight that altering the steepness of individual drug dose-response curves may offer under-appreciated opportunities for drug design.

The predictions of our model are directly testable in experiments (shown above and in Ref. [12]). Perhaps the most striking experimental validation of our model is the change in drug interaction type due to the presence of antibiotic resistance genes (Figs 9D and 10C). This observation is notable since previous work concluded that most mutations and mechanisms that provide resistance to individual drugs only rescale the effective antibiotic concentrations while preserving the shape of the dose-response curves and surfaces [30–33]. In contrast, our results show that specific resistance genes for two antibiotics targeting the ribosome inevitably alter the drug interaction, even in the absence of more complicated mechanisms.

Discrepancies between experimental results and model predictions can expose cases in which more complicated mechanisms cause the observed drug interaction [12]. A limitation of our model is that it considers fully assembled translating ribosomes as sole targets of the antibiotics, without taking the exact stage of the translation cycle into account. In principle, a model that describes ribosome assembly and more details of the translation cycle and the transitions between its different steps could provide a more detailed mechanistic picture. However, since we currently do not know the *in vivo* parameter values that characterize the translation cycle, such a model would not be predictive, but would rather rely on extensive fitting of free parameters to limited experimental data. Instead, the underlying mechanisms of drug interactions that cannot be captured by the minimal biophysical model presented here, and in particular suppression, can be elucidated by targeted phenomenological approaches [12].

## Supporting information

**S1 Appendix. Additional model analysis, numerical methods, and experimental considerations.**
(PDF)

## Acknowledgments

B.K. is thankful to C. Guet for additional guidance and generous support which rendered this work possible. We thank all members of Guet, Bollenbach, and Tkačik groups for many helpful discussions and sharing of laboratory resources. B.K. additionally acknowledges the tremendous support from A. Angermayr and K. Mitosch with experimental work. We thank M. Hennessey-Wesen and M. Zagórski for constructive comments on the manuscript.

## Author Contributions

**Conceptualization:** Bor Kavčič, Gašper Tkačik, Tobias Bollenbach.

**Data curation:** Bor Kavčič.

**Formal analysis:** Bor Kavčič.

**Funding acquisition:** Bor Kavčič, Gašper Tkačik, Tobias Bollenbach.

**Investigation:** Bor Kavčič, Gašper Tkačik, Tobias Bollenbach.

**Methodology:** Bor Kavčič, Gašper Tkačik, Tobias Bollenbach.

**Project administration:** Gašper Tkačik, Tobias Bollenbach.

**Resources:** Gašper Tkačik, Tobias Bollenbach.

**Software:** Bor Kavčič.

**Supervision:** Gašper Tkačik, Tobias Bollenbach.

**Validation:** Bor Kavčič, Gašper Tkačik, Tobias Bollenbach.

**Visualization:** Bor Kavčič.

**Writing – original draft:** Bor Kavčič, Gašper Tkačik, Tobias Bollenbach.

**Writing – review & editing:** Bor Kavčič, Gašper Tkačik, Tobias Bollenbach.

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
