## [Decision Letter · Decision Letter 0]

18 Sep 2020

Dear Dr. Bollenbach,

Thank you very much for submitting your manuscript "Minimal biophysical model of combined antibiotic action" for consideration at PLOS Computational Biology. As with all papers reviewed by the journal, your manuscript was reviewed by members of the editorial board and by several independent reviewers. The reviewers appreciated the attention to an important topic. Based on the reviews, we are likely to accept this manuscript for publication, providing that you modify the manuscript according to the review recommendations.

Both the reviewers found the paper interesting, correct, and timely. They included a list of suggestions for minor modifications. In addition, as suggested by the second reviewer, it is essential that the authors clearly explain why they have not tested experimentally some of the predictions.

Sincerely,

Jacopo Grilli

Associate Editor

PLOS Computational Biology

Feilim Mac Gabhann

Editor-in-Chief

PLOS Computational Biology

[LINK]

Both the reviewers found the paper interesting, correct, and timely. They included a list of suggestions for minor modifications. In addition, as suggested by the second reviewer, it is essential that the authors clearly explain why they have not tested experimentally some of the predictions.

Reviewer's Responses to Questions

**Comments to the Authors:**

Reviewer #1: This is an interesting and significant manuscript which presents and analyses a minimal model for the combined action of pairs of ribosome targeting antibiotics on a bacterial cell. Understanding the combined action of antibiotics is crucially important since many clinical therapies involve combinations of antibiotics, and it is well known that interactions can be either synergistic or antagonistic, or even suppressive. However, there are few quantitative predictive models for drug interactions (or even for the action of individual antibiotics). Therefore this work makes an important contribution to the field. The work presented is careful, thorough, mostly clearly described, and supported by some original experimental data. I support its publication in PLoS Computational Biology, although I have some suggestions for the authors to consider.

Suggestions:

1. The caption of Fig 2 should cite Scott et al (2010) [9] and Greulich et al (2015) [10].

2. It would be useful to give the expression for the bound ribosome concentration rb before Eq (2), since that is needed in Eq (7).

3. On line 94, ref [10] is not appropriate (just ref. [9])

4. There are blank spaces in place of section numbers in several places, eg before Eq 6, line 305, line 480.

5. Lines 151-153 state that “in general the parameters delta can vary continuously to capture any change in ribosome binding of one antibiotic due to binding of the second” – I don’t think this is quite true. For example if the binding becomes non-linear this would not be captured?

6. In lines 159-163 the wording is confusing where it refers to “bound to one antibiotic” and “single-bound antibiotic” – here we are anyway only talking about the case of single binding?

7. Before Eq (7) the parameter xi appears with no explanation, and in Eq (7) the expression for rb = Delta r(1-lambda/lambda0) appears without explanation – this should have been explained explicitly before Eq (2). Also it should be made clear that the final relation applies only along the isobole.

8. In Eq (8) it should be made clear this in the steady state. Also after Eq (8) it was not clear to me what is meant by the proportionality constant.

9. Eq (8) is not actually an equation for the isobole (that would be an equation that links the two antibiotic concentrations for fixed growth rate). I think the statement “Eq (8) corresponds to a linear isobole” would be clearer if an actual equation for the isobole can be stated.

10. Lines 171-173 seemed a little out of place and perhaps distracting.

11. Lines 187-188 a sentence or two to describe Fig 3c in more detail might be useful for clarity.

12. Lines 189-196: the wording here implies a time-line (antibiotic 1 binds before antibiotic 2) that is not accurate: the model does not specify which antibiotic binds first. The explanation here could be rewritten to reflect this.

13. Lines 203-204 the wording is a little hard to follow, e.g. “growth is inhibited close to zero” – what is the assumption here, that the antibiotic concentration is low so there is little inhibition?

14. In the caption of Fig 4b, it states that the dashed line is for antibiotics with different alpha values. But the axis label seems to imply the alpha values are the same.

15. Lines 225-230 the statement “it is unclear if mutual stabilization of binding necessarily leads to synergy” is not backed up by the text that follows. Actually it seems to imply that mutual stabilization of binding does lead to synergy; in contrast, it is if the two antibiotics mutually destabilize that the interaction is no longer synergistic.

16. Lines 324-326 – this statement that the model can be generalized to other modes of antibiotic action is not clear to me. The model is based around the growth rate-dependent regulation of ribosome abundance. It is not obvious to me that it can easily be generalized to other modes of action. I feel that this statement is interesting but deserves a more in-depth justification, for example in the discussion section.

17. Lines 329-330 The statement that the growth rate is invariant of the target details as it can be recovered from first order kinetics is unclear to me. What is meant by first order kinetics here? In my view the core of the model is specific to ribosome-binding antibiotics.

18. Line 377 onwards: it seems important to clarify here that the experiments were done with E. coli (rather than S. aureus which is mentioned early in the paragraph).

19. Lines 396-7 “recent work” – it is perhaps not that recent (2013).

20. Line 428 The word “drastic” here is perhaps overkill since the previous sentence states that the effect is “slight”.

21. The result that the presence of resistance genes can qualitatively change the nature of the drug interaction is very nice and could perhaps be made more prominent in the abstract (it is there but is rather buried under other things).

22. I also wonder whether how this phenomenon (effect of resistance genes) depends on the growth rate?

Reviewer #2: The manuscript studies a mathematical model for the combined action of two or more growth-inhibiting antibiotics. In these combinations at least one of the studied antibiotics is a translation-inhibiting antibiotic therefore the authors base their model on a previous model (Greulich et al, 2015) for the mode of action of a translation-inhibiting antibiotic, which takes into account the coupling between antibiotic action and bacterial growth.

The authors predict the interaction of antibiotics (antagonistic, independent, or synergistic) for different combinations of antibiotic types. In particular, for antibiotics which simultaneously bind to the ribosome, they study how binding interactions between antibiotics on the same ribosome (e.g. of allosteric origin) affect the combined drug efficacy. In that scenario, the combined effect is explicitly modelled as an extension of the model (Greulich et al. 2015), and it is predicted that the combined drug effects -- antagonistic or synergistic -- are enhanced if drug molecule binding is synergistic, and weakened if drug molecule binding is antagonistic. This is surprising when considering the naive view that synergistic drug binding should generally lead to a synergistic drug effect. This is an interesting prediction worth to be tested experimentally.

It is further studied how translation-inhibiting antibiotics interact with starvation-mimicking antibiotics and how the presence of resistance genes affects the drug interaction. Those predictions are also tested experimentally in bacterial growth assays where antibiotics are applied and genetically modified bacterial strains are used to study the effect of resistance genes.

This study is a very interesting and solid piece of work, with a thorough, comprehensive study of a realistic model (partially confirmed in the past) and experimentally testable predictions. Some of these predictions are directly tested experimentally in this work. Predictions about the effect and interaction of antibiotics (e.g. dose-response curves) as done in this work are of high medical value.

The only concern I have is that the authors decided to test some of their predictions experimentally, but others not. While it is generally a plus of this study that model predictions are experimentally tested, there is the risk of a confirmation- or selective bias if only some predictions are experimentally tested. It is therefore essential that the authors give a clear rationale why other predictions have not been tested experimentally, and argue thoroughly that no selection bias occurs. Of course, if possible, doing the corresponding experiments for those predictions would be preferable and strengthen this work even further.

Further minor comments:

1. line 105: For completeness, it should be said that the form if s(lambda) stated here is only valid in the steady state (the steady state is only mentioned in the following sentence).

2. below Eq. 6, index are sometimes named "A" and "B" and sometimes "1" and "2". It would be good to be consistent in the notation here.

3. line 199: it is stated that large alpha means that binding is more reversible, however, it could also mean that antibiotic outflux of the cell is large.

4. The authors prefer using Loewe's definition of drug interactions (antagonistic/independent/synergistic). Please argue why this definition is preferred, since in the following it is shown that this crucial classification depends sensitively on this definition. In particular, for the limit studied in Eq. (10), the interaction is synergistic in Loewe's view, but independent in Bliss' view, so the terms "synergistic" and "independent" are just semantics at this point.

5. line 323: It is stated that Eq. 16 is consistent with Eq. 14. This does not become clear at this point. Could the authors show this at this point or refer to an Appendix?

6. In Fig. 7c there are substantial deviations between experimental data and model predictions for streptomycin. Please comment on this deviation.

7. At several points in the text, references to Appendix 5 are made for experimental details, but these are actually given in Appendix 6.

**Have all data underlying the figures and results presented in the manuscript been provided?**

Reviewer #1: Yes

Reviewer #2: **No: **Tables of experimental data missing. If not done already, the code for numerical compuations should be published.

PLOS authors have the option to publish the peer review history of their article (what does this mean?). If published, this will include your full peer review and any attached files.

Reviewer #1: No

Reviewer #2: No
---

## [Decision Letter · Decision Letter 1]

12 Nov 2020

Dear Dr. Bollenbach,

We are pleased to inform you that your manuscript 'Minimal biophysical model of combined antibiotic action' has been provisionally accepted for publication in PLOS Computational Biology.

Best regards,

Jacopo Grilli

Associate Editor

PLOS Computational Biology

Feilim Mac Gabhann

Editor-in-Chief

PLOS Computational Biology

As suggested by reviewer 1, please make sure that the experimental data are available.

Reviewer's Responses to Questions

**Comments to the Authors:**

Reviewer #1: The authors have addressed all my comments and I am happy to recommend publication.

Reviewer #2: The authors have addressed all my remarks satisfactorily. They explain well their choice of experiments and why the seen deviations between experiments and data are to be expected.

The authors state that "All relevant data are within the manuscript and its Supporting Information files", however, I cannot find the numerical values of the data. I do understand that all data is shown in the figures, but for reproducibility it would be good to have the experimental data also in numerical form, e.g. in a data table.

**Have all data underlying the figures and results presented in the manuscript been provided?**

Reviewer #1: Yes

Reviewer #2: **No: **Experimental data is shown in plots but not numerically. Numerical values of the data should be shown or referenced if they are in an external data base.

PLOS authors have the option to publish the peer review history of their article (what does this mean?). If published, this will include your full peer review and any attached files.

Reviewer #1: No

Reviewer #2: No

---

## [Editor Report · Acceptance letter]

18 Dec 2020

PCOMPBIOL-D-20-01159R1 

Minimal biophysical model of combined antibiotic action

Dear Dr Bollenbach,

I am pleased to inform you that your manuscript has been formally accepted for publication in PLOS Computational Biology. Your manuscript is now with our production department and you will be notified of the publication date in due course.

With kind regards,

Livia Horvath
